# Antifouling Systems Based on Copper and Silver Nanoparticles Supported on Silica, Titania, and Silica/Titania Mixed Oxides

**DOI:** 10.3390/nano12142371

**Published:** 2022-07-11

**Authors:** Carla Calabrese, Valeria La Parola, Simone Cappello, Annamaria Visco, Cristina Scolaro, Leonarda Francesca Liotta

**Affiliations:** 1Istituto per lo Studio dei Materiali Nanostrutturati (ISMN)-CNR, Via Ugo La Malfa 153, I-90146 Palermo, Italy; carla.calabrese@ismn.cnr.it (C.C.); valeria.laparola@cnr.it (V.L.P.); 2Institute for Biological Resources and Marine Biotechnologies (IRBIM), National Research Council (CNR), Spianata San Raineri 86, I-98122 Messina, Italy; simone.cappello@cnr.it; 3Department of Engineering, University of Messina, Contrada Di Dio, I-98166 Messina, Italy; annamaria.visco@unime.it (A.V.); cristina.scolaro@unime.it (C.S.); 4Institute for Polymers, Composites and Biomaterials, CNR-IPCB, via P. Gaifami 18, I-95126 Catania, Italy

**Keywords:** marine biofouling, silver nanoparticles, copper nanoparticles, silica, titania, mixed oxides

## Abstract

Silica, titania, and mixed silica–titania powders have been used as supports for loading 5 wt% Cu, 5 wt% Ag, and 2.5 wt% Cu-2.5 wt% Ag with the aim of providing a series of nanomaterials with antifouling properties. All the solids were easily prepared by the wetness-impregnation method from commercially available chemical precursors. The resulting materials were characterized by several techniques such as X-ray diffraction analysis, X-ray photoelectron spectroscopy, N_2_ physisorption, and temperature-programmed reduction measurements. Four selected Cu and Ag SiO_2_- and TiO_2_-supported powders were tested as fillers for the preparation of marine antifouling coatings and complex viscosity measurements. Titania-based coatings showed better adhesion than silica-based coatings and the commercial topcoat. The addition of fillers enhances the resin viscosity, suggesting better workability of titania-based coatings than silica-based ones. The ecotoxicological performance of the powders was evaluated by Microtox luminescence tests, using the marine luminescent bacterium *Vibrio fisheri*. Further investigations of the microbiological activity of such materials were carried out focusing on the bacterial growth of *Pseudoalteromonas* sp., *Alteromonas* sp., and *Pseudomonas* sp. through measurements of optical density at 600 nm (OD_600nm_).

## 1. Introduction

Marine biofouling is perceived as a concern by both the scientific community and industrial parties owing to the environmental and economic impacts of shipping via the aquatic ecosystem. In 2011, the International Maritime Organization (IMO) published the Resolution MEPC.207(62) with the aim of providing guidelines for the control and management of biofouling by ships to minimize the transfer of invasive aquatic species. In principle, biofouling can be defined as a series of adhesion and deposition phenomena of living matter on a given solid surface that is exposed to aqueous media or, at least, to some moisture [1]. In particular, marine biofouling originates from the adhesion of marine organisms (e.g., bacteria, algae, barnacles) to any solid surface submerged in seawater [2]. In this context, the proliferation of fouling organisms on ship hulls increases the surface roughness hampering the overall drag performance. The consequential fuel consumption, which is required to maintain a specific speed, allows higher air emissions including greenhouse gases (GHG), sulfur oxides (SO_x_), nitrogen oxides (NO_x_), particulate matter (PM), volatile organic compounds (VOC), and ozone-depleting substances [3]. Moreover, fouling organisms may travel with ships they stick to and spread to sea areas far from their habitats, becoming biological invaders and affecting local ecosystems.

The need to search for viable solutions to tackle marine biofouling issues has allowed the development of antifouling and anticorrosion coatings associated with the design of nanocomposite materials based on biocidal agents or acting through the so-called fouling release mechanism [4]. From a general point of view, the development of nanocomposite materials with antimicrobial and antifouling properties represents a well-known topic finding application in many research fields including the fabrication of biomedical devices, water purification systems, and food packaging as well as marine equipment [5,6,7,8].

In this perspective, metal and metal oxide nanoparticles (NPs) have been studied as antimicrobial agents which, due to their nanosized structure and subsequent high surface-to-volume ratio, display enhanced bioactivity in terms of microbial cell damage [9,10,11]. In this framework, the antimicrobial performance of metal nanoparticles is strongly related to their dispersion and stabilization on a given medium able to prevent agglomeration phenomena that are detrimental to the overall biocidal activity.

Recently, some of us have reviewed the antifouling and antimicrobial activity of Ag and Cu nanoparticles supported on silica and titania [12]. The choice of silica and titania as metal supports is based on their tunable surface properties (e.g., high specific surface area, hydrophobic/hydrophilic balance) combined with other intrinsic parameters such as high thermal, chemical, and biological stability. Moreover, the specific Brønsted and Lewis acidity sites of titanium dioxide promote the stabilization of metallic nanoparticles. In the field of materials science, TiO_2_-based nanocomposites subjected to light excitation are remarkably effective in eliciting microbial death. Upon photoactivation of the oxide component, the biocidal action is a result of the modulation of charge (electron-hole) carriers at the interface of the external surface of the material [13].

When reporting on the design of nanomaterials, it is crucial to consider the importance of the applicative footprint together with the environmental and cost impacts of the adopted synthetic protocols. Based on this, we propose a series of antimicrobial materials made up of Ag- and Cu-NPs on silica, titania, and silica–titania mixed oxide as solid supports. Copper and silver-based materials were easily prepared via the wetness-impregnation method over commercial SiO_2_ and TiO_2_ powders. Moreover, silica and titania have been mechanically mixed in order to obtain additional support with synergistic features, such as a high specific surface area, from SiO_2_ combined with TiO_2_ to stabilize its capability against metal nanoparticle sintering. The obtained materials were characterized by nitrogen physisorption, X-ray photoelectron spectroscopy (XPS), X-ray diffraction (XRD), and temperature-programmed reduction (TPR). Selected materials were tested as fillers for the preparation of marine antifouling coatings that were deposited over a steel metal support (DH34 steel). The viscosity of the commercial resin added to the biocidal agents has been examined and adhesion cross-cut tests of the deposited coatings have been performed.

In parallel, the ecotoxicological assay of the powders was performed by Microtox luminescence tests using the marine bacterium *Vibrio fisheri*. Further investigations on the microbiological activity of such materials were carried out by monitoring the bacterial growth of *Pseudoalteromonas* sp., *Alteromonas* sp., and *Pseudomonas* sp. through measure-of-growth density (OD_600nm_) measurements.

## 2. Materials and Methods

### 2.1. Samples Preparation

All chemicals used for the synthesis were provided by Sigma Aldrich with 99.99% purity and were used without any further purification. The commercial silica was purchased from Merck (amorphous silica gel 60).

Copper and silver-based materials were prepared through wetness-impregnation method of the precursors onto the support oxides. Cu(NO_3_)_2_ 2.5H_2_O and AgNO_3_ were used as metal precursors, whereas commercial SiO_2_, TiO_2_, and SiO_2_–TiO_2_ (1:1) mixed powders were selected as the solid supports. In particular, the SiO_2_–TiO_2_ (1:1) support was obtained by mechanical mixing of SiO_2_ and TiO_2_ oxides. Copper and silver precursors, in appropriate amount to loading 5 wt% Cu, 5 wt% Ag and 2.5 wt% Cu-2.5 wt% Ag, were dissolved in the minimum volume of distilled water and added drop by drop to the support. Then, after drying at 120 °C overnight, the materials were calcined at 500 °C for 2 h with a heating ramp of 5 °C/min.

A representation of the synthesis and composition of the Cu/Ag-based materials is given in Figure 1.

### 2.2. Samples Characterization

X-ray diffraction measurements of Cu- and Ag-containing materials were performed on a Bruker D 5000 diffractometer equipped with a Cu Kα anode in the range of 20 to 80° (2θ) with a step size of 0.05° and a time per step of 20 s. The crystalline phase composition of the powders was analyzed according to the ICSD database (FIZ Karlsruhe, Leibniz Institute, 2022 release, Eggenstein-Leopoldshafen, Germany). The mean crystallite size of crystalline phases was calculated by the Debye–Scherrer equation: D = 0.9 λ/Bcosθ, where D represents the average crystalline size, 0.9 is the Scherrer parameter, λ is the wavelength of the X-ray radiation (0.15406 nm), B denotes the full width at half the maximum of the peak (FWHM), and θ is the angular position of the main peak.

Specific surface area (SSA), pore volume, and mean pore diameter of the materials were determined by N_2_ physisorption analysis at −196 °C using ASAP 2020 Plus instrument (Micromeritics, Norcross, GA, United States). Prior to the analysis, the samples were outgassed at 200 °C under a vacuum for 2 h. The Brunauer−Emmett−Teller (BET) method was used to calculate the SSA. The Barrett−Joyner−Halenda (BJH) method was applied to the desorption branch to calculate the mesoporous pore volume and the average pore diameter.

The X-ray photoelectron spectroscopy (XPS) analyses of the powders were carried out using a VG Microtech ESCA 3000 Multilab (VG Scientific, Sussex, UK) with an Al Kα source (1486.6 eV) run at 14 kV and 15 mA in CAE(Constant Analyser Energy) mode. For the individual peak energy regions, pass energy of 20 eV set across the hemispheres was used. The constant charging of the samples was removed by referencing all the energies to the C 1 s peak energy set at 285.1 eV, arising from adventitious carbon. The XPS data were examined using the CASA XPS software (version 2.3.17, Casa Software Ltd. Wilmslow, Cheshire, UK, 2009). For the peak shape, a Gaussian (70%)-Lorentzian (30%) line shape, defined in Casa XPS as GL (30) profiles, was used for each component of the main peaks after a Shirley background subtraction. The binding energy values were quoted with a precision of ±0.15 eV and the atomic percentage with a precision of ±10%.

The materials were examined by temperature-programmed reduction (H_2_-TPR) using Micromeritics Autochem 2910 apparatus equipped with a thermal conductivity detector (TCD). The H_2_-TPR profiles were registered after pre-treatment of the solids under 5 vol% O_2_/He flow (30 mL/min) from room temperature up to 200 °C, holding for 30 min. Then the samples were cooled down to room temperature. The H_2_-TPR analysis was carried out in the temperature range of 25–1000 °C with a 10 °C/min heating rate by flowing a gas mixture of 5 vol% H_2_/Ar (30 mL/min).

### 2.3. Coatings Preparation

The powder fillers synthesized as described in Section 2.1 were mechanically dispersed individually within a resin based on a commercial polymeric matrix (Hempel’s Silic One, HEMPEL S.R.L Genova, Italy topcoat, blue-colored) [14], to obtain a loading of 0.1 weight %. This resin is based on silicone and is biocide-free; its low friction surface prevents organisms from attaching to the surface that are in contact with the seawater [15]. The prepared topcoats with the various synthetic antifouling fillers were then deposited on a metal support (DH34 steel) and pre-treated with a white-colored layer of primer (Hempel’s Light Primer, HEMPEL S.R.L Genova, Italy ~116 μm thick) and a second yellow-colored layer of tie-coat (Hempel’s Silic One, HEMPEL S.R.L Genova, Italy ~180 μm thick); both are commercial reagents and have the same chemical composition as the topcoat (see Figure 1).

We have coded each coating as “HSMX”: H represents the commercial Hempel resin (biocide-free) for the finish coating, SMX is the type of filler we synthesized and added by mixing, and X specifies its type (i.e., X = 1, 5, 6, 7) according to Table 1. Pure topcoat resin, H, was used as a reference sample.

### 2.4. Coatings Characterization

The adhesion of coating films to the metallic DH36 steel substrate was evaluated by a cross-cut test, performed using a commercial Cross Hatch Adhesion Tester (SAMA Tools SADT502-5, SAMA Italia, Viareggio, Italy) according to the ASTM D3359e2 Standard Test Method for Measuring Adhesion by Tape Test. Using an appropriate cutter, a grid incision was made in a test area of approximately 10 × 10 cm, creating a grid of horizontally (2 mm) and vertically (2 mm) spaced incisions across the surface [16]. All the particles produced in the area were then removed with a soft brush. As a rule, a 3M adhesive tape is stuck onto the cutting grid with a finger applying light pressure. It is subsequently removed with an even peeling movement. The test is evaluated visually by comparing the sectional grid image with the reference images from ISO 2409:2013. Depending on the condition of the damage, a cross-cut parameter from ISO-0 (very good adhesive strength) to ISO-5 (very poor adhesive strength) is assigned according to the number of squares that have flaked off as well as the appearance.

The workability of liquid-like samples at room temperature was analyzed through a rotational rheometer (MC-502, Anton Paar, Graz, Austria) consisting of a rotating rod immersed in the coating to be analyzed. The terminal part of the rod has a specific shape (geometry). Measurements of complex viscosity were carried out using parallel flat-plate geometry at room temperature. For each geometry, the lower support was always fixed (only the upper part is connected to the motor). A dynamic stress sweep test (frequency of 1 Hz) was performed within the stress range of 0.5–1000 Pa to check the linear viscoelastic region (LVR). From this test, we chose a stress value of 1.3 Pa within the LVR to test our coatings. Thus, frequency sweep tests (in stress control) were carried out in the frequency range 0.01–100 rad/s at a constant stress value of 1.3 Pa. Each test was carried out three times.

### 2.5. Microtox Assay

The Microtox^®^ luminescence assay was performed on Cu- and Ag-based materials in order to detect any toxic substances that were released in the water by the above-mentioned biocidal agents. Microtox^®^ toxicity tests were conducted using the luminescent bacterium Vibrio fisheri according to the standard procedures of the EN12457 protocol with the following modifications. Toxicity values (bioluminescence inhibition) were reported as the effective concentrations of toxicants resulting in a 50% decrease in bioluminescence (EC50). EC50 with 95% confidence intervals was calculated following the procedures outlined in the Microtox^®^ System Operating Manual (Microtox, Columbus, Ohio, 2003). Each biocidal sample was compared with a reference un-toxicity matrix.

### 2.6. Bacteria, Culture Conditions, and Bacteriostatic Activity Tests of Biocidal Agents

In order to evaluate the bacteriostatic activity of the synthesized biocidal agents as marine antifoulings, three strains of *Pseudoalteromonas sp., Alteromonas sp., and Pseudomonas* sp. were used in all the experiments. The strains used in this study belong to a collection of marine bacteria held at IRBIM-CNR of Messina.

Starter cultures were carried out by inoculating one loop of microbial cells into 10 mL of Marine Broth (Difco, Milan) mineral medium. Cultures were grown in a rotary shaker (New Brunswick C24KC, Edison NJ, USA; 150 rpm) at 20 ± 1 °C for 5 days. Mid-exponential-phase grown cells were harvested by centrifugation at 11.250× *g* for 10 min, washed twice with sterile medium, and inoculated into different 100 mL sterile Erlenmeyer flasks each containing 50 mL of Marine Broth. Mid-exponential-phase grown cells were harvested a second time as previously described and inoculated at a final concentration of 0.1 of optical density (OD600nm) in a sterile medium added with 100 mgL-1 of Cu- and Ag-based materials. The cultures were incubated (for 9 days) in a rotary shaker (New Brunswick C24KC, Edison NJ, USA; 150 rpm) at 20 ± 1 °C. All experiments were carried out in triplicate. At the begging of experimentation and at regular intervals (3 days), the growth (biomass variations) of the cultures in the study was routinely assessed indirectly by measuring the turbidity (OD600nm) using a UV–visible spectrophotometer (Shimadzu UV-160, Markham, ON, Canada). All the experiments were conducted in triplicate.

## 3. Results and Discussion

### 3.1. Samples Characterization

The crystalline structure of the samples was analyzed by XRD measurements. In Figure 2, the XRD patterns of the prepared materials are displayed along with the ICSD reference files for CuO, Cu_2_O, Ag_2_O, and Ag.

For the silica-based materials SM1 and SM2, the XRD patterns show a broad peak at around 2θ = 22° due to amorphous SiO_2_ and two main intense peaks at 2θ = 35.8° and 38.8° along with secondary features at 48.9°, 61.6°, 66.2°, and 68.1°, related to CuO crystallites (ICSD n. 16025). The mean crystallite size of the CuO phase, estimated from Scherrer’s equation applied to the most intense peak, was equal to 9 and 6 nm for SM1 and SM2, respectively. Very-low-intensity peaks related to CuO were also identified on the supported SiO_2_-TiO_2_ samples, SM3 and SM4, and on the TiO_2_-based samples, SM5 and SM6. No signals attributable to Cu_2_O (ICSD n. 26963) were detected.

The main reflections of TiO_2_ anatase (ICSD n. 9852) can be identified as well as a small amount of rutile phase (ICSD n. 9161). Interestingly the copper-containing catalysts show a decrease in the rutile phase with respect to anatase, as evidenced by the decrease in the peaks at 27.5°, 26,1°, and 54.3°. With respect to the detection of silver, no diffraction peaks of metallic Ag (ICSD n. 64706) or Ag_2_O (ICSD n. 281041) species were observed. On that basis, the presence of highly dispersed silver phases with a size below the detection limit of the XRD technique (≤3.0 nm) was inferred, although we cannot exclude the presence of some cationic Ag species strongly interacting with the TiO_2_ support or that the relatively low Ag loading could limit the detection of crystalline phases.

N_2_ physisorption analysis was performed in order to evaluate the surface area, pore volume, and pore-size distribution of the synthesized samples. The isotherms are classical type IV, as defined by IUPAC, with hysteresis typical of mesoporous materials, especially for the silica-based ones. According to the specific surface area of bare silica (320 m^2^/g), the SM1 and SM2 materials showed values around 300 m^2^g^−1^, pore volumes around 0.68 cm^3^g^−1^, and a relatively narrow pore size distribution centered at 6.9 nm. Cu, Ag, and Cu-Ag TiO_2_ exhibited slightly lower specific surface areas (45–48 m^2^g^−1^) with respect to the TiO_2_ bare support (56 m^2^g^−1^) and pore volume values around 0.47 cm^3^g^−1^.

The deposition of Cu and Cu-Ag on the mechanical mixture SiO_2_–TiO_2_ allowed obtaining the materials SM3 and SM4 with intermediate values of the specific surface area (170–175 m^2^g^−1^) and pore volume (0.53–0.55 cm^3^g^−1^) compared to the high surface area of SiO_2_ and the relatively low surface area of TiO_2_.

The reduction properties of the solids were investigated by temperature-programmed reduction analysis (H_2_-TPR). The TPR profiles are reported in Figure 3. The temperatures at the maxima of the reduction peaks and the H_2_ consumption values are listed in Table 2. The TPR profile of 5Cu/SiO_2_ exhibits a main peak centered at 241 °C, which is attributed to the reduction of the CuO nanoparticles with a relatively big size according to the mean crystal size of 9 nm calculated by Scherrer’s equation, not interacting with the support. No hydrogen consumption can be ascribed to the SiO_2_. In the case of the 5Cu/SiO_2_–TiO_2_ and 5Cu/TiO_2_ materials, two additional small peaks at 153 and 167 °C were detected along with the main hydrogen consumption at around 241 °C. The low-temperature peaks were attributed to the presence of easily reducible highly dispersed CuO nanoparticles. In all cases, the full reduction of Cu^2+^ to metallic Cu is intended to occur in only one step, although we cannot rule out that a two-step reduction, Cu^2+^→Cu^+^ at low temperatures and Cu^+^→Cu^0^ at higher temperatures, takes place [17,18]. Overall, the total hydrogen consumption measured for SM1, SM3, and SM5 is in good agreement with the theoretical values expected for the full reduction of Cu^2+^ to metallic Cu. A low contribution from the reduction of surface TiO_2_ (that, in the presence of Cu-Ag, occurs at around 250 °C) and from the reduction of bulk TiO_2_ (at around 975 °C) must also be taken into account. In this respect, TPR analysis of the bare TiO_2_ evidences a total hydrogen consumption of 2.6 mL/g.

In the silver-containing samples, hydrogen consumption in the range ~65–140 °C was associated with a reduction of oxidized silver nanoparticles and clusters of silver oxides with different sizes [19,20]. The reduction profiles of Cu-Ag bimetallic materials are characterized by a main peak centered at 229 or 280 °C, ascribed to the reduction of CuO species interacting differently with the support; the higher the interaction, the higher the reduction temperature. Reduction features below 200 °C were attributed to the reduction of oxidized silver and/or copper species. As seen in the results in Table 2, for samples SM2, SM4, SM6, SM7, and SM8, the experimental hydrogen consumption is in good agreement with the theoretical one expected for the full reduction of Cu^2+^ and or Ag+ to the metallic species. On the other hand, for the titania-supported materials, we must take into account the occurrence of some surface and bulk reduction of the support that contributes to the overall experimental hydrogen consumption.

The surface composition of the materials was studied using X-ray photoelectron spectroscopy. The oxygen signal is typical of the one relative to the support with a peak at 532.5 eV for silica and two peaks at 528.6 and 531.7 eV for titania; the chemical mixture showed a mixture of signals due to the two pure oxides.

The Cu2p region of 2.5Cu-2.5Ag/SiO_2_ is shown in Figure 4a as a representative of all samples and the Cu2p3/2 binding energy values are listed in Table 3. The peak position and peak shape are typical of CuO for all the samples [21,22]. For all the samples, silver shows the typical Ag3d doublet at 368.3 and 374.3 eV for Ag 3d572 and Ag3d3/2, respectively (See Figure 4b). The positions of the silver 3d peaks can be attributed to metallic silver produced by the thermal decomposition of Ag_2_O at 400 °C [23,24]. The possible formation of metallic Ag nanoparticles agrees well with the antibacterial activity registered for the Ag-based materials, which is in agreement with the antifouling and antimicrobial properties reported in the literature for the supported silver nanoparticles [12,25]. However, the presence of residual Ag_2_O cannot be ruled out because of the closeness of the position related to the two species.

As expected for well-dispersed supported metals, the signals of surface Cu and Ag over SiO_2_–TiO_2_ are higher with respect to the theoretical values for the silica-supported samples and increase for the pure titania. Interestingly, the highest Cu surface amount was found for 5Cu/TiO_2_ (SM5), which resulted in the most active biocidal material. A similar Ag surface enrichment was observed in the monometallic silica–titania and pure titania-supported samples in comparison with the silica-supported ones. In the bimetallic materials, the Ag/Cu ratio showed a surface enriched with silver for the titania-based samples.

### 3.2. Evaluation of the Adhesion Power of the Coatings and Their Rheological Features

The evaluation of the adhesive properties of the coating films deposited over the metallic DH36 steel substrate was carried out through a cross-cut test. The coatings made in the laboratory by mixing the antifouling fillers at 0.1 wt% with the commercial silicone resin H gave an improvement in terms of the adhesive properties compared to the reference resin H.

The results of the cross-cut test are shown in Figure 5 for the two selected samples, HSM1 and HSM5, in comparison with the commercial topcoat H. We can see that for coating H (Figure 5a) and the silica-based HSM1 (Figure 5b), the edges of the cuts are not completely flat because in some parts the squares of the lattice are detached. In accordance with the reference standards, we have an adhesion evaluated as 3B (specifically for the ASTM D 3359-09e2) and ISO-2 (specifically for the ISO 2409:2007).

Instead, the titania-based HSM5 coating is better in adhesion compared to the other two coatings (Figure 5c). The detachment of small varnish lamellae occurred at the intersections of the cuts. Thus, this unique high scratch resistance proves the good adhesion levels of these coatings, typical of 4B adhesion (specifically for the ASTM D 3359-09e2) and ISO-1 (specifically for the ISO 2409:2007).

These results showed that the titania-based coating (HSM5) has better adhesion in comparison with the silica-based one (HSM1) as well as the commercial topcoat.

In Figure 6, we can observe the trend of the complex viscosity (η *) as a function of the frequency (from 0.01 rad/s to 100 rad/s) for the four samples, HSM1, HSM5, HSM6, and HSM7. The reference resin H shows the lowest viscosity out of all the coatings. At 1 rad/s, its complex viscosity is ~1000 rad/s. The addition of fillers enhances the resin viscosity as expected. The silica-based sample, HSM1, has the highest viscosity value across the frequency range and was analyzed at 1 rad/s, its complex viscosity is ~275,000 rad/s. The group of titania-based samples is positioned at intermediate levels of viscosity: the HSM5 sample seems to coincide with the viscosity value of the HSM7 sample at high-frequency values, from 1 rad/s to 100 rad/s, confirming the similar characteristics of these two materials. At 1 rad/s, the complex viscosity of HSM5 and HSM7 is ~24,000 rad/s. These results suggested to us that the titania-based coatings have better workability than the silica-based ones thanks to their lower viscosity.

### 3.3. Ecotoxicological Assays (Microtox Toxicity Tests)

Following the EN12457 protocol, we combined the Cu- and Ag-based materials with sterile water at both 1:2 and 1:10 ratios (wt/vol) and no significant level of bioluminescence decay was observed, ruling out the release of toxic compounds by the biocidal agents employed. As reported in the EN12457 protocol, Microtox^®^ bioluminescent assay tested in water typically exhibits an underestimated sensitivity against highly hydrophobic chemicals. This is mainly due to both the extremely low solubility of these compounds in water and the almost irreversible adsorption to a matrix similar to sediments.

### 3.4. Antibacterial Activity Tests

The growth curves of bacterial strains in the study were obtained by determining the time-dependent growth of optical density (OD_600nm_). Measurements were registered in the presence of five selected biocidal agents, SM1, SM2, SM5, SM6, and SM7, and compared with a blank test carried out with the H resin, biocide-free.

The time-dependent changes in the abundance of bacterial growth were monitored for 9 days at the beginning of the experiment and then every 3 days (Figure 7). The best antibacterial activity was found for the 5Cu/TiO_2_; the silica-supported solids (5Cu/SiO_2_ and 2.5Cu-2.5Ag/SiO_2_) showed medium-low antibacterial activity, whereas the titania-supported Cu-Ag and Ag materials exhibited intermediate antibacterial activity between SM2 and SM5. The blank test carried out with the H resin, biocide-free, confirmed higher bacterial growth than that observed in the presence of the biocidal agents.

These results show that the use of titania allowed obtaining biocides with improved antibacterial properties with respect to the silica-based ones. The best performance of the 5Cu/TiO_2_ powder was ascribed to the presence of well-dispersed CuO nanoparticles, as revealed by XRD, TPR, and XPS characterizations, as well as to a synergistic effect between the copper species and the titanium dioxide.

It is likely that the mutual interaction between TiO_2_ and the cationic Cu species strongly interacting with the support, gives rise to active sites that have improved antibacterial properties, as seen in a comparison of the Cu supported on SiO_2_, and Cu-Ag- and Ag-supported on TiO_2_ oxides. 

## 4. Conclusions

A series of nanomaterials based on copper and silver nanoparticles have been easily prepared by the wetness-impregnation of silica, titania, and mechanically mixed silica–titania powders. The obtained materials were characterized by X-ray diffraction (XRD) analysis, N_2_ physisorption, temperature-programmed reduction (TPR) analysis, and X-ray photoelectron spectroscopy (XPS). Four selected samples based on Cu and supported SiO_2_ and TiO_2_ powders were used as fillers for the preparation of polymeric coatings to evaluate their viability as efficient systems to mitigate marine fouling phenomena. Titania-based coatings showed better adhesion than the silica-based ones, and a commercial topcoat was used as a reference. The addition of fillers enhanced the resin viscosity, with the Cu–silica sample showing the highest viscosity value and it was also the most difficult to work with. Conversely, the Cu and Ag titania-based coatings exhibited good workability thanks to their relatively low viscosity. The ecotoxicological performance of the materials was evaluated by Microtox toxicity tests using the marine luminescent bacterium *Vibrio fisheri*. Any release of toxic substances was excluded. The microbiology (bacteriostatic and/or bacteriocide) activity was carried out by analysis of the bacterial growth of three selected marine strains (*Pseudoalteromonas* sp., *Alteromonas* sp., *Pseudomonas* sp.) in the presence of five selected Cu and Ag biocides. A blank test carried out with the H resin, biocide-free, confirmed higher bacterial growth than that observed in the presence of the biocidal agents. In general, the bacterial strains under investigation showed a unique behavior. The best performance identified as an inhibition of bacteria in the study, was fulfilled by the Cu/TiO_2_ powder that was characterized by well-dispersed CuO species in a synergistic interaction with titania.

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
