# Peer review of "Antifouling Systems Based on Copper and Silver Nanoparticles Supported on Silica, Titania, and Silica/Titania Mixed Oxides"

_nanomaterials, 2022, doi:10.3390/nano12142371_

Round 1

Reviewer 1 Report

-          At Introduction, specify the advantages of chose method.

-          Schematic representation of obtained materials must be added.

-          At 2.1, specify the all chemicals used in this study (line 96, “All the other chemicals”)

-          Please, specify the applications or sectors of the industry were these materials can be used.

-          - English improvement is required. Some examples but not all are as the following:

        - At Introduction, line 64, modify “(e.g. high specific surface area” with “(e.g., high specific surface area”

        - At 3, line 232, modify “180-280°C” with “180-280 °C”

        - At 3, line 238, modify “1.9 ml/g” with “1.9 mL/g”

Author Response

Dear reviewer, please, enclosed you will find the responses. 

Reviewer 2 Report

Title: Antifouling systems based on silver and copper nanoparticles supported on silica, titania and silica/titania mixed oxides 

Manuscript ID: nanomaterials-1798801

The authors report antifouling coatings based on silver and copper nanoparticles prepared by the wetness impregnation. There is no doubt that this paper has many new discoveries, but some problems need to be revised. The author should make further revision to the article.

General comments: 

1. Line 198-203, references should be cited to support the conclusion.

2. Line 201-203, the description “silver particles with a size below the detection limit of the XRD technique” can not be obtained based on the fact of “no diffraction peaks of metallic Ag or Ag2O species were observed”. Other characterization methods should be used to verify the existence of Ag or Ag2O and their sizes. For example, the scanning electron microscope energy-dispersive X-ray spectroscopy analysis should be provided to confirm the desperation of Ag or Ag2O.

3. According to the Introduction, this coating seems to be applied in marine biofouling. Is Microtox toxicity testing sufficient to assess antimicrobial performance? Explanation should be provided.

4. How about the durability of the coating? Corrosion test should be provided, such as neutral salt spray test.

Author Response

(The authors gave the same response as above.)

Reviewer 3 Report

The paper "Antifouling systems based on silver and copper nanoparticles supported on silica, titania and silica/titania mixed oxides" presents interesting and original results on the fabrication of the new antifouling coatings. The paper can be published in Nanomaterials mdpi after major revision.

It will be very valuable to provide any information about synthesized silver and copper nanoparticles supported on silica, titania and silica/titania mixed oxides. What about their size, shape, or charge?

In the antibacterial activity tests is absent information on the antibacterial properties of the fillers without antibacterial nanoparticles. This information is valuable to compare the differences between samples. 

Please add an appropriate discussion about the antibacterial properties of the silver and copper nanoparticles supported on silica, titania, and silica/titania mixed oxides.

Finally, I suggest to cite the following references where similar results were presented.

https://doi.org/10.1016/j.colsurfa.2022.128525

10.1039/C6TB00051G

Author Response

(The authors gave the same response as above.)

Round 2

Reviewer 1 Report

Dear Sirs,

The manuscript was improved and it can be publish in this form.

Reviewer 2 Report

This manuscript was revised detailedly according to the comments of reviewers, and the quality is much improved. I’m glad to recommend publication.

Reviewer 3 Report

After revision the quality of the paper was essentially improved and paper can be accepted in the present form.